# ON THE NECESSITY OF LEARNABLE SHEAF LAPLACIANS

**Ferran Hernandez Caralt, Mar Gonzàlez i Català, Pietro Liò**
Department of Computer Science and Technology
University of Cambridge
Cambridge, United Kingdom
`{fh455,mg2211,pl219}@cam.ac.uk`

**Adrián Bazaga**
Microsoft
`adrianbazaga@microsoft.com`

## ABSTRACT

Sheaf Neural Networks (SNNs) were introduced as an extension of Graph Convolutional Networks to address oversmoothing on heterophilous graphs by attaching a sheaf to the input graph and replacing the adjacency-based operator with a sheaf Laplacian defined by (learnable) restriction maps. Prior work motivates this design through theoretical properties of sheaf diffusion and the kernel of the sheaf Laplacian, suggesting that suitable non-identity restriction maps can avoid representations converging to constants across connected components. Since oversmoothing can also be mitigated through residual connections and normalization, we revisit a trivial sheaf construction to ask whether the additional complexity of learning restriction maps is necessary. We introduce an Identity Sheaf Network baseline, where all restriction maps are fixed to the identity, and use it to ablate the empirical improvements reported by sheaf-learning architectures. Across five popular heterophilic benchmarks, the identity baseline achieves comparable performance to a range of SNN variants. Finally, we introduce the Rayleigh quotient as a normalized measure for comparing oversmoothing across models and show that, in trained networks, the behavior predicted by the diffusion-based analysis of SNNs is not reflected empirically. In particular, Identity Sheaf Networks do not appear to suffer more significant oversmoothing than their SNN counterparts.

## 1 INTRODUCTION

Oversmoothing and heterophily are two phenomena in graph neural networks (GNNs) that have been widely discussed, yet remain loosely defined in the literature Wang et al. (2024); Bodnar et al. (2022). In the context of node classification, oversmoothing refers to node representations becoming overly similar, making it difficult to distinguish classes Bodnar et al. (2022). A graph is said to be heterophilous Wang et al. (2024) when edges tend to connect nodes from different classes Wang et al. (2024). Both issues have been linked to the aggregation operator used by many GNNs, such as the adjacency-based message-passing mechanism Bodnar et al. (2022); Cai & Wang (2020).

Motivated by this connection, SNNs were introduced to mitigate these problems by replacing the adjacency operator with a sheaf Laplacian Bodnar et al. (2022). Following Bodnar et al. (2022), several variants have been proposed and evaluated on the same benchmark suites Barbero et al. (2022); Zaghen et al. (2024); Suk et al. (2022); Caralt et al. (2024); Ribeiro et al. (2026); Fiorini et al. (2026), often accompanied by theoretical analyses centered on properties of the Sheaf Laplacian. Nevertheless, Scholkemper et al. (2025) shows that residual connections and normalization can be sufficient to mitigate oversmoothing. This raises a natural question: is the additional complexity of the Sheaf Laplacian necessary in practice? To address it, we revisit the trivial Sheaf Laplacian construction introduced by Hernández Caralt (2024) and use it as a baseline to benchmark the empirical improvements attributed to learned sheaf constructions.

**Contributions.** We show that a trivial sheaf Laplacian, obtained by fixing all restriction maps to the identity, achieves competitive performance across all heterophily benchmarks used by Bodnar et al. (2022), and we relate this observation to the heterophily characterization of Wang et al. (2024). We also introduce the Rayleigh quotient as a normalized measure to quantify oversmoothing and enable comparisons across different models. Using this measure, we show that the diffusion-based theoretical intuition commonly used to motivate SNNs is not borne out empirically in trained networks, suggesting that future work should reconsider which theoretical framework best explains their practical behavior.

## 2 BACKGROUND

**Notation.** We will consider a graph $G = (V, E)$, with $V = \{1, 2, ..., n\}$ where each node $u \in V$ has some features $x_u \in \mathbb{R}^f$ attached to it. We will denote with $X \in \mathbb{R}^{n \times f}$, the matrix where its $i$-th row is $x_i^T$. $A$ will be the adjacency matrix of the graph and $D$ a diagonal matrix with $D_{ii} = \deg(i)$. $\Delta_G = I - D^{-1/2}AD^{1/2}$ will be the graph laplacian of $G$. $\sigma$ will denote a non-linear activation.

With this notation in mind, the first Graph Neural Network that was introduced by Kipf & Welling (2016) and is still state of the art across many benchmarks Luo et al. (2025) consists of the following.

**Definition 2.1.** Given a graph $G$ with node features $X$, a **Graph Convolutional Network (GCN)** is defined layer-wise with $H^{(0)} = X$ and $H^{(t+1)} = \sigma \left( D^{-1/2}AD^{1/2}H^{(l)}W_t \right)$

Bodnar et al. (2022) claimed that this architecture suffered from oversmoothing because stacking layers in a GCN could be seen as an Euler discretisation of $\frac{dX(t)}{dt} = -\Delta_G X(t)$, namely taking $X(t+1) = X(t) - \Delta_G X(t) = (I - \Delta_G)X(t) = D^{-1/2}AD^{1/2}X(t)$. This is related to oversmoothing through the following property shown by Bodnar et al. (2022).

**Proposition 2.2.** *The solution to the ODE $\frac{dX(t)}{dt} = \Delta_G X(t)$ satisfies $\lim_{t \to \infty} X(t) \in \ker(\Delta_G) = \{x_u = x_v | (u, v) \in E\}$. In the limit, the solution is constant across connected components.*

Proposition 2.2 implied that stacking layers of a GCN made representations converge to constants in connected components. To break away from this, sheaves were proposed by Bodnar et al. (2022).

**Definition 2.3.** Given $G = (V, E)$, a **cellular sheaf** $\mathcal{F}$ on $G$ consists of: (i) for each $v \in V$ a set $\mathcal{F}(v)$ which is called the **stalk** of $v$, and idem for $e \in E$, (ii) for each pair $v \trianglelefteq e$ a linear map $\mathcal{F}_{v \trianglelefteq e} : \mathcal{F}(v) \longrightarrow \mathcal{F}(e)$, usually called **restriction map**.

From this concept one can define a Sheaf Laplacian that is not limited by Proposition 2.2:

**Definition 2.4.** Given a graph $G$ and a sheaf $\mathcal{F}$ on it, its **sheaf laplacian** is defined as $\Delta_\mathcal{F} = \delta_\mathcal{F}^T \delta_\mathcal{F}$ where the operator $\delta_\mathcal{F} : \bigoplus_{v \in V} \mathcal{F}(v) \to \bigoplus_{e \in E} \mathcal{F}(e)$ is called the coboundary map and is defined like: $(\delta_\mathcal{F} x)_e = \mathcal{F}_{u \trianglelefteq e} x_u - \mathcal{F}_{v \trianglelefteq e} x_v$. The image of this laplacian at each node is $(\Delta_\mathcal{F} x)_u = \sum_{e \in E | u \trianglelefteq e} \mathcal{F}_{u \trianglelefteq e}^T (\mathcal{F}_{u \trianglelefteq e} x_u - \mathcal{F}_{v \trianglelefteq e} x_v)$.

**Proposition 2.5.** *If we consider the sheaf diffusion $\frac{dX(t)}{dt} = -\Delta_\mathcal{F} X(t)$, then $\lim_{t \to \infty} X(t) \in \ker(\Delta_\mathcal{F}) = \{\mathcal{F}_{u \trianglelefteq e} x_u = \mathcal{F}_{v \trianglelefteq e} x_v | (u, v) = e \in E\}$.*

In other words, if the right sheaf $\mathcal{F}$ was computed, it would be possible to stack many layers without representations converging to a constant. In fact, all of their theoretical results on the linear separation power of Sheaf Neural Networks relied on the kernel of the Sheaf Laplacian, and the convergence of Sheaf Diffusion to said kernel. With this motivation, Bodnar et al. (2022) proposed a generalization of GCNs using this sheaf laplacian.

**Definition 2.6.** Let us denote $X(t) \in \mathbb{R}^{nd} \times \mathbb{R}^f$, where $n$ is the number of nodes, $d$ is the dimension of the stalks, $f$ the number of feature channels, and $t$ the layer of the neural network. The update equation of a Sheaf Neural Network would be $X^{(t+1)} = X^{(t)} - \sigma(\Delta_{\mathcal{F}^{(t)}}(I_n \otimes W_1^{(t)} X^{(t)} W_2^{(t)})$ where the restriction maps of $\mathcal{F}^{(t)}$ are computed like Bodnar et al. (2022): $\mathcal{F}_{u \trianglelefteq e}^{(t)} = MLP(x_u^{(t)} || x_v^{(t)})$ where $v \trianglelefteq e$, and there are two hyperparameters that may fix one diagonal element of all the terms $\mathcal{F}_{u \trianglelefteq e}^T \mathcal{F}_{v \trianglelefteq e}$ to $+1$ and $-1$ respectively.

Other works by Barbero et al. (2022); Zaghen et al. (2024); Suk et al. (2022); Caralt et al. (2024); Ribeiro et al. (2026); Fiorini et al. (2026) introduced various changes to this architecture: defining

the maps at preprocessing time, a nonlinear version of the laplacian, wave propagation, further establishing the relevance of Sheaf Neural Networks and directional and cooperative extensions.

## 3   IDENTITY SHEAF NETWORKS

In this section we propose the model introduced by Hernández Caralt (2024) as a baseline to ablate the use of sheaves in graph neural networks. This will consist of considering a fixed trivial sheaf.

**Definition 3.1.** Given a graph $G = (V, E)$, an **Identity Sheaf** is a sheaf $\mathcal{F}$ satisfying $\mathcal{F}_{u \trianglelefteq e} = \mathcal{F}_{v \trianglelefteq e} = I_d$ for all $(u, v) \in E$. We will denote this type of sheaves with $\mathcal{I}$.

**Definition 3.2.** An **Identity Sheaf Network** (ISN) consists of a Sheaf Neural Network, where the learnable sheaf is an Identity Sheaf $\mathcal{I}$. This architecture is equivalent to a GIN Xu et al. (2018) except for the way the convolutions $W_1^{(t)}, W_2^{(t)}$ are done. This change can be seen as a sparser linear layer.

Given that all of the theoretical motivation behind sheaves relies on non-identity restriction maps Bodnar et al. (2022); Zaghen et al. (2024); Barbero et al. (2022); Caralt et al. (2024); Ribeiro et al. (2026); Fiorini et al. (2026) comparing these results with ISN should properly ablate the sheaf learning methods. Results for the original benchmarks used by Bodnar et al. (2022)[1] can be seen in Table 1, highlighted in green are the results of other sheaf works where the increase in performance of SNNs with respect to ISN is greater than standard deviation. As we can observe, ISN achieves comparable results to all SNNs.

| Model | Texas | Wisconsin | Squirrel | Chameleon | Cornell |
|---|---|---|---|---|---|
| ISN | $88.01 \pm 4.05$ | $88.82 \pm 3.83$ | $53.27 \pm 2.28$ | $67.35 \pm 1.71$ | $85.95 \pm 5.38$ |
| Best-RiSNN | $87.89\pm4.28$ ($-0.12\pm4.05$)$\approx$ | $88.04\pm2.39$ ($-0.78\pm3.83$)$\approx$ | $53.30\pm3.30$ ($+0.03\pm2.28$)$\approx$ | $66.58\pm1.81$ ($-0.77\pm1.71$)$\approx$ | $85.95\pm6.14$ ($0.00\pm5.38$)$\approx$ |
| Best-jDSNN | $87.37\pm5.10$ ($-0.64\pm4.05$)$\approx$ | $89.22\pm3.42$ ($+0.40\pm3.83$)$\approx$ | $51.28\pm1.80$ ($-1.99\pm2.28$)$\approx$ | $66.45\pm3.46$ ($-0.90\pm1.71$)$\approx$ | $85.41\pm4.55$ ($-0.54\pm5.38$)$\approx$ |
| Conn-NSD | $86.16\pm2.24$ ($-1.85\pm4.05$)$\approx$ | $88.73\pm4.47$ ($-0.09\pm3.83$)$\approx$ | $45.19\pm1.57$ ($-8.08\pm2.28$)$\downarrow$ | $65.21\pm2.04$ ($-2.14\pm1.71$)$\downarrow$ | $85.95\pm7.72$ ($0.00\pm5.38$)$\approx$ |
| Best-SNN | $85.95\pm5.51$ ($-2.06\pm4.05$)$\approx$ | $89.41\pm4.74$ ($+0.59\pm3.83$)$\approx$ | $56.34\pm1.32$ ($+3.07\pm2.28$)$\uparrow$ | $68.68\pm1.73$ ($+1.33\pm1.71$)$\approx$ | $86.49\pm7.35$ ($+0.54\pm5.38$)$\approx$ |
| Best-NSP | $87.03\pm5.51$ ($-0.98\pm4.05$)$\approx$ | $89.02\pm3.84$ ($+0.20\pm3.83$)$\approx$ | $50.11\pm2.03$ ($-3.16\pm2.28$)$\downarrow$ | $62.85\pm1.98$ ($-4.50\pm1.71$)$\downarrow$ | $76.49\pm5.28$ ($-9.46\pm5.38$)$\downarrow$ |
| Best-NLSD | $87.57\pm5.43$ ($-0.44\pm4.05$)$\approx$ | $89.41\pm2.66$ ($+0.59\pm3.83$)$\approx$ | $54.62\pm2.82$ ($+1.35\pm2.28$)$\approx$ | $66.54\pm1.05$ ($-0.81\pm1.71$)$\approx$ | $87.30\pm6.74$ ($+1.35\pm5.38$)$\approx$ |
| Best-DSNN | $88.65\pm4.95$ ($+0.64\pm4.05$)$\approx$ | $90.20\pm4.02$ ($+1.38\pm3.83$)$\approx$ | NA | NA | $87.84\pm5.70$ ($+1.89\pm5.38$)$\approx$ |
| Best-CSNN | $87.30\pm5.93$ ($-0.71\pm4.05$)$\approx$ | $90.00\pm2.83$ ($+1.18\pm3.83$)$\approx$ | NA | NA | $81.62\pm4.32$ ($-4.33\pm5.38$)$\approx$ |

Table 1: Results across 5 popular heterophilic benchmarks of ISN against the results shown by Bodnar et al. (2022); Caralt et al. (2024); Suk et al. (2022); Barbero et al. (2022); Zaghen et al. (2024); Ribeiro et al. (2026); Fiorini et al. (2026). Main numbers are $\mu \pm \sigma$. The colored annotations denote the difference vs. ISN, as $\Delta\mu \pm \sigma$ (green: improvement, red: degradation, gray: no change).

## 4   HETEROPHILY ANALYSIS

We use the heterophily measure proposed by Wang et al. (2024) to explain why the ISN achieves performance comparable to SNNs on the benchmarks reported in Table 1. In particular, Wang et al. (2024) argues that heterophily can be categorized as *bad*, *mixed*, or *good* based on the probability distributions of node neighborhoods. More specifically, they define:

**Definition 4.1.** Given $G = (V, E)$, where each node has a label attached $y_v \in \{1, ..., c\}$, $\hat{m}_k \in \mathbb{R}^c$ for $k \in \{1, ..., c\}$ a vector where each entry $\hat{m}_{ki}$ represents the proportion of nodes of class $i$ that are in the neighborhood of class $k$ and $d_k$ the average degree of nodes of class $k$. Then the **gain** between classes $k$ and $t$ is defined as $\text{gain}(k, t) = ||\sqrt{d_k}\hat{m}_k - \sqrt{d_t}\hat{m}_t||$

---

[1]Results on the dataset film are included in the Appendix due to reproducibility issues encountered.

**Definition 4.2.** Given $G = (V, E)$ where each node has a label attached $y_v$, for $\varsigma = 0.2$[2]it is said to have good heterophily if $\min_{k \neq t}$ gain(k,t) $> \varsigma$, **bad** if $\max_{k \neq t}$ gain(k,t) $< \varsigma$, and **mixed** otherwise.

If we apply this measure to the benchmarks we obtain the results shown in Table 2. As we can see, all datasets show good heterophily patterns, thus explaining why IdentitySheaf, which is equivalent to a GCN Bodnar et al. (2022), can achieve comparable results to SNNs.

| Dataset | Texas | Wisconsin | Squirrel | Chameleon | Cornell |
|---|---|---|---|---|---|
| Min Gain | 0.89 | 0.98 | 0.64 | 0.57 | 0.54 |
| Max Gain | 3.98 | 4.3 | 2.99 | 3.67 | 3.1 |
| Type of Heterophily | Good | Good | Good | Good | Good |

Table 2: Heterophily measures introduced by Wang et al. (2024) on the benchmarks used.

## 5 OVERSMOOTHING ANALYSIS

The proposed analysis for why sheaves prevent oversmoothing done by Bodnar et al. (2022) focused on Proposition 2.5. This means that, in sheaf diffusion, connected components are not necessarily constant in the limit, enabling non-smooth representations of the data when stacking many layers.

This analysis was also extended by Bodnar et al. (2022) to the Dirichlet Energy $X^T \Delta_G X = \sum_{(u,v) \in E} (x_u - x_v)^2 \to 0$ arguing that regular diffusion minimizes this quantity while sheaf diffusion minimizes $X^T \Delta_{\mathcal{F}} X = \sum_{(u,v) \in E} ||\mathcal{F}_{u \trianglelefteq e} x_u - \mathcal{F}_{v \trianglelefteq e} x_v||^2 \to 0$, again enabling adjacent nodes to converge to different representations. Thus, we propose to test the following hypothesis

**Hypothesis 5.1.** Given a SNN's representation $X_{\mathcal{F}}(t)$ and an ISN's representation $X_{\mathcal{I}}(t)$, then $X_{\mathcal{F}}^T \Delta_{\mathcal{F}} X_{\mathcal{F}} \to 0$ and $X_{\mathcal{F}}^T \Delta_{\mathcal{I}} X_{\mathcal{F}} \not\to 0$, while $X_{\mathcal{I}}^T \Delta_{\mathcal{I}} X_{\mathcal{I}} \to 0$.

To compare these across different models, we use the following as a normalized Dirichlet Energy.

**Definition 5.2.** Given a semi-definite positive matrix $\Delta$, its **Rayleigh Quotient** is $R_\Delta(x) = \frac{x^T \Delta x}{x^T x}$

More specifically, we compute $R_{\Delta_{\mathcal{F}}}(x(t))$ and $R_{\Delta_{\mathcal{I}}}(x(t))$ for the representations $x(t)$ at each layer of a trained SNN and an ISN. In Figure 1 we see that the differences in the sheaf space are generally higher than in the regular space, implying that the trained models contradict the Hypothesis 5.1. Furthermore, ISNs do not appear to suffer from more significant oversmoothing than their SNN counterparts[3].

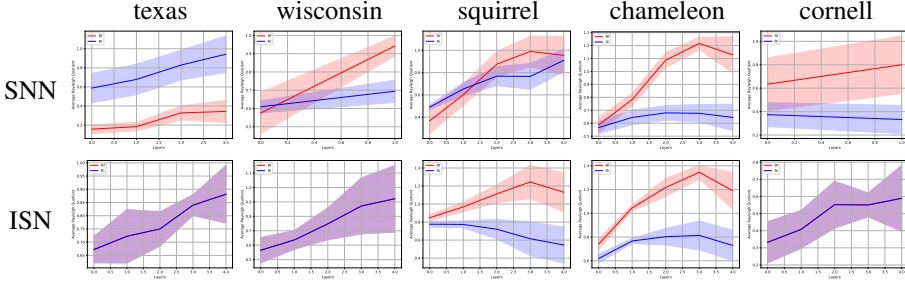

Figure 1: A series of line plots, one for each dataset-neural network combination, representing the averaged (across folds) $R_{\Delta_{\mathcal{F}}}$ in red and $R_{\Delta_{\mathcal{I}}}$ in blue at each layer. These results contradict the theoretical understanding proposed by Bodnar et al. (2022) since according to Hypothesis 5.1, the blue line should be above the red line.

---

[2]The theoretical definition given by Wang et al. (2024) used $\varsigma = 1$, however in the practical use cases they used $\varsigma = 0.2$ to account for correlations in the graph, which is the value we will also use.

[3]Note that in Squirrel and Chameleon, ISN uses the hyperparameter described in Defintion 2.6 that alters $\mathcal{F}_{u \trianglelefteq e}^T \mathcal{F}_{v \trianglelefteq e}$, thus why $R_{\Delta_{\mathcal{F}}} \neq R_{\Delta_{\mathcal{I}}}$.

## 6 CONCLUSIONS AND FUTURE WORK

The main takeaways of this work are that learnable restriction maps are not necessary to mitigate oversmoothing on the standard benchmarks studied here, and that the diffusion-based analysis of oversmoothing via the (sheaf) Laplacian is not currently supported by the empirical behavior we observe in trained models. Consequently, our results discourage interpreting the sheaf Laplacian primarily through the lens of a diffusion equation and its kernel as the space to which representations converge. As future work, it would be interesting to carry out a similar analysis for the recent approach of Bamberger et al. (2025), and to evaluate the remaining datasets used by Ribeiro et al. (2026); Fiorini et al. (2026) once their code is released (or becomes available) so that the reported results can be reproduced.

### ACKNOWLEDGEMENTS

Ferran Hernandez Caralt acknowledges that the project that gave rise to these results received the support of a fellowship from "la Caixa" Foundation (ID 100010434). The fellowship code is LCF/BQ/PFA25/11000012. Mar Gonzàlez i Català acknowledges G-Research's support for the development of this work.

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

## A    APPENDIX: RESULTS ON FILM DATASET

For the Film dataset, the best result that could be reproduced was 36.35±0.95 for SNN (instead of the 37.81±1.15 shown by Bodnar et al. (2022)) and 36.61±0.86 for ISN. The min gain was 0.56 and the max gain 3.30, making it a dataset with good heterophily. The Rayleigh quotient plot can be seen in 2, which is different as the best model contained only a single layer. These results are aligned with the other ones discussed in this paper.

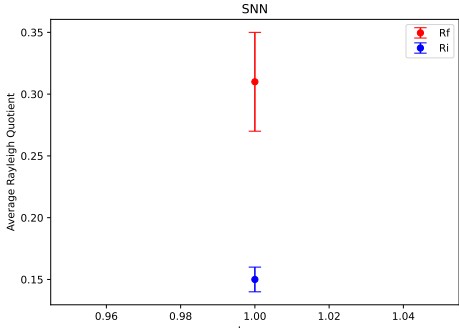 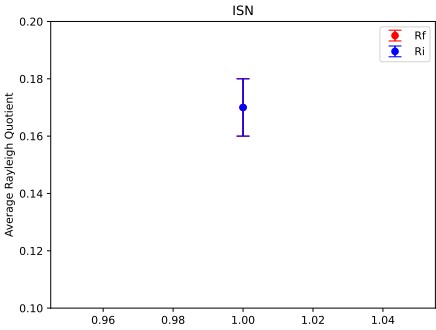

Figure 2: Rayleigh Quotients at the first and only layer of a trained SNN and ISN. $R_{\Delta_{\mathcal{F}}}$ in red and $R_{\Delta_{\mathcal{I}}}$ in blue at each layer.

## B    APPENDIX: CODE

The code can be found in `https://github.com/ferranhernandezc/baseline-sheaf-diffusion`.

## C    APPENDIX: HYPERPARAMETER CONFIGURATIONS

The hyperparameter configurations for the SNN trained are the ones provided by Bodnar et al. (2022) in their code. For ISNs the ones in Table 3 were used.

| | Texas | Wisconsin | Film | Squirrel | Chameleon | Cornell |
|---|---|---|---|---|---|---|
| add_hp | False | True | False | False | False | False |
| add_lp | False | False | False | True | True | False |
| d | 1 | 1 | 4 | 1 | 1 | 1 |
| deg_normalised | False | False | True | False | False | False |
| dropout | 0.7 | 0.8 | 0.5 | 0 | 0 | 0.9 |
| early_stopping | 200 | 200 | 100 | 100 | 100 | 200 |
| epochs | 1500 | 500 | 1000 | 1000 | 1000 | 500 |
| hidden_channels | 32 | 96 | 64 | 96 | 32 | 64 |
| input_dropout | 0 | 0 | 0.1 | 0.7 | 0.7 | 0 |
| layers | 5 | 5 | 1 | 5 | 5 | 4 |
| lr | 0.03 | 0.02 | 0.0002 | 0.01 | 0.01 | 0.02 |
| normalised | True | True | False | True | True | True |
| second_linear | False | False | True | True | True | False |
| weight_decay | 0.005 | 0.0006685729356 | 0.0000001 | 0.0001121579137 | 0.0002969905682 | 0.0006914841723 |

Table 3: Hyperparameter configurations of the ISNs that were used in this work.

