# OpenReview forum: "On the necessity of learnable sheaf laplacians"
_ICLR.cc/2026/Workshop/GRaM — ICLR 2026 Workshop GRaM Poster_

### Official Review · Reviewer_6Eme · 2026-02-17
**Mischaracterization of sheaf Dirichlet energy weakens the oversmoothing analysis**

**Rating:** 4
**Confidence:** 4

**Review:**

**Summary:** This paper questions whether learnable restriction maps in Sheaf Neural Networks (SNNs) are necessary for their reported empirical gains on heterophilic benchmarks. The authors introduce an Identity Sheaf Network (ISN) baseline where all restriction maps are fixed to the identity matrix, and show it achieves comparable performance to various SNN variants.

**Strengths**
1. Clear and well-written
2. Since sheaves have a strong geometric flavor and avoiding the learning of too many restriction maps helps SNNs scale, the topic is relevant to GRaM.
3. The research question "are learnable sheaves necessary?" is valuable.
4. Investigating if SNNs are really avoiding over-smoothing is important.

**Weaknesses**
1. The central analysis (Section 5) mischaracterizes Bodnar et al.'s claims about Dirichlet energy (I will expand this below).
2. The benchmarks are small and exhibit high variance, which is acceptable for an extended abstract but makes the conclusion that learnable restriction maps are unnecessary too strong.
3. Does not cite Sheaf Attention Networks (Barbero et al), which more directly address the oversmoothing issue.

Minor: Missing year in the reference for Keyulu Xu et al.

**General comment:** I agree that the experiments in Bodnar et al. do not address the oversmoothing problem thoroughly and that the relevance of each learned map is an important open question. However, I am concerned that this extended abstract could lead to further misunderstanding of the role of the sheaf Dirichlet energy in Bodnar et al.'s work.

The author say "The proposed analysis for why sheaves prevent oversmoothing done by Bodnar et al. (2022) focused on Proposition 2.5", where Proposition 2.5 refers to the sheaf diffusion. While it is true that sheaf diffusion is used to argue about the expressive power of sheaf-based models, the argument is only that sheaf convolutional neural networks (not necessarily the NSD model specifically) are more expressive than GCNs in the sense that they are generally not constrained to decrease the Dirichlet energy when using low-norm weights. Crucially, there is no suggestion in Bodnar et al. that the sheaf Dirichlet energy should itself be used as a measurement of oversmoothing. The key idea is that the sheaf Dirichlet energy measures oversmoothing when the sheaf is trivial, but for any other sheaf it captures something else. Consequently, the Rayleigh quotient analysis would be more meaningful if applied with the standard normalized graph Laplacian, so that only the node representations change across models.

The results in Bodnar et al that tackle oversmoothing are those in the section "The Linear Separation Power of Sheaf Diffusion", which  show that under certain circumstances there are sheaves with linear separation power, i.e., sheaves that can properly solve a node classification task.  The motivation behind learning the sheaf is then to find one with such separation power. Thus, results in Table 1 could indicate that SNNs struggle to learn the "right" sheaf rather than indicating it is not necessary to learn it.

I encourage the authors to investigate further for which datasets it is necessary to learn the sheaf and to look for evidence of when and why sheaves can mitigate oversmoothing (I recommend checking Sheaf Attention Networks by Barbero et al. on this point). However, I do not recommend this work for acceptance in its current form.

**Clarification based on comment of Reviewer nyTE:** I do not deny the connection established between sheaf Dirichlet energies and expressive power of SNNs. My point is that Bodnar et al. (2022) do not connect the sheaf Dirichlet energy with oversmoothing. The connection with oversmoothing is done in another section. The authors of this tiny paper, however, use their finding that sheaf Dirichlet energy is not connected with oversmoothing to argue that this contradicts Bodnar et al. (2022). But the contradiction does not exist, since no such claim was made in the original work. I feel it is important to flag this. The authors' empirical findings are valuable on their own and would be strengthened if presented as a standalone contribution rather than framed as a refutation.

**Pmlr Suitability:**

NA

---

### Official Review · Reviewer_nyTE · 2026-02-19
**Important empirical finding on the (lack of) necessity for non-trivial restriction maps**

**Rating:** 7
**Confidence:** 4

**Review:**

**Summary:**:

This paper investigates whether learnable restriction maps within Sheaf Neural Networks (SNNs) are indeed necessary to improve performance and prevent oversmoothing in graph neural networks.
The authors compare against a simple baseline called an Identity Sheaf Network (ISN) ( Hernandez Caralt (2024)), which keeps the same architecture as SNNs but fixes all restriction maps to the identity instead of learning them. Across five standard heterophilic graph benchmarks, this simpler model performs about as well as existing SNN variants. This suggests that the extra complexity of learning sheaf maps may not be needed.


**Strengths:**
1) The considered research question is timely and valuable.
2) The (negative) results are an important reality check.
3) The paper is well written.

**Weaknesses:**
The considered datasets are all small. While it should be acknowledged that graph learning in general struggles with a suboptimal dataset situation, there are nevertheless bigger and newer heterophilic graph datasets available.
For any future conference level submission of this work, substantial additional numerical verifications (on such newer and bigger heterophilic datasets) are necessary.

**Comment on the analysis of Reviewer 6Eme:**
The main issue the reviewer finds with the current submission seems to be that the present submission ''[...]mischaracterizes Bodnar et al.'s claims about Dirichlet energy". In the reviewer's eyes, Bodnar et al.'s 2022 paper does not link Dirichlet energy to separation capabilities, so that the Dirichlet energy analysis of the present submission (Section 5) seemingly becomes superfluous. However, that is not the case. In Bodnar et al (2022) it is for examply explicilty written (c.f. page 6 ibid.), that

"[...] we investigate how SCN layers affect the sheaf Dirichlet energy [...]"
Then Arguments relating Dirichlet energy to Kernels of Sheaf Laplacians are presented and finally it is concluded that
"Therefore, in the linear separation capabilities of [models with symmetric restriction maps] are
severely limited ."

Hence here (and in additional examples (c.f. also the summary-box on page 7 of Bodnar et. al. (2022) )) There is indeed a connection established between Dirichlet energies, Kernels of Sheaf Laplacians and expressive power.

Hence, to me it strongly seems that the criticism of Reviewer 6Eme is infact not applicable, and the Dirichlet Energy analysis presented in the current submission is both valid and in line with the literature.


**Decision:**  The negative result discovered (i.e. that trivial restriction maps perform as well as learned ones) is an important reality check with regards to the applicability and understanding of sheaf networks.
**The paper has substantially more than sufficient merit to be accepted in the tiny paper track.**

**Pmlr Suitability:**

NA

---

### Meta-Review · Area_Chair_Mr1n · 2026-02-27

**Decision:**

Accept

**Metareview:**

There was some disagreement between reviewers on the merits of this paper. I concur with reviewer nyTE that the criticism by 6Eme might be misplaced. Since the paper is in a tiny paper track, I think it could be useful to be presented at the workshop. I would advise the authors to use suggestions by both the reviewers to improve their discussion (including of prior work) and clarity.

**Relevance To Proceedings:**

Tiny paper — does not apply

**Relevance To Workshop:**

Yes — suitable for GRaM

---

### Decision · Program_Chairs · 2026-03-02

Accept (Poster)